# Limited application of reflective surfaces can mitigate urban heat pollution

Sushobhan Sen [1] & Lev Khazanovich[1]

Elevated air temperatures in urban neighborhoods due to the Urban Heat Island effect is a form of heat pollution that causes thermal discomfort, higher energy consumption, and deteriorating public health. Mitigation measures can be expensive, with the need to maximize benefits from limited resources. Here we show that significant mitigation can be achieved through a limited application of reflective surfaces. We use a Computational Fluid Dynamics model to resolve the air temperature within a prototypical neighborhood for different wind directions, building configurations, and partial application of reflective surfaces. While reflective surfaces mitigate heat pollution, their effectiveness relative to cost varies with spatial distribution. Although downstream parts experience the highest heat pollution, applying reflective surfaces to the upstream part has a disproportionately higher benefit relative to cost than applying them downstream.

[1] Department of Civil and Environmental Engineering, Swanson School of Engineering, University of Pittsburgh, Pittsburgh, PA, USA. ✉email: sus128@pitt.edu

A large body of literature over the past 200 years has shown that urban areas tend to have higher air temperatures than adjacent rural areas[1,2], the so-called Urban Heat Island (UHI) effect. Kleerekoper et al.[3] summarized the causes of the development of UHI, including the replacement of vegetative surfaces with impervious ones of lower solar reflectance (albedo), which absorb and retain greater solar radiation, as well as a reduction of turbulent heat transport due to dissipation of wind as a result of urban form. Manoli et al.[4] also found that the UHI is strongly influenced by the absence of permeable surfaces and moisture. These factors are typical of dense urban areas, where over half the world's population lives[5], a proportion that is expected to rise steadily in the coming decades. The UHI effect is particularly concerning because it is a form of heat pollution that gradually corrodes quality of life and the environment. Previous studies have shown that this heat pollution leads to increased consumption of electricity for cooling buildings[6–8], higher water consumption[9,10], reduced thermal comfort[11–13], and deterioration in public health[14–16]. Ziter et al.[17] showed that these effects also varied within a city and correlated with the extent of impervious services at local scales. Thus, it is important for urban planners and policymakers to consider solutions for heat mitigation. However, with limited budgets that need to be stretched to provide municipal services, there is a need for solutions and strategies that maximize the benefits from the limited resources available to communities to mitigate heat pollution.

Several interventions to mitigate heat pollution have been recommended in the literature. Green infrastructure, which refers to the creation of new green spaces as well as covering existing surfaces with vegetation in the form of green roofs and green walls, is one such measure. It is based on the effect of latent heat from evapo-transpiration as well as shading to decrease the air temperature. Balany et al.[18], Saaroni et al.[19], and Tzoulas et al.[20] reviewed a number of studies that demonstrated the benefits of green infrastructure as a heat pollution mitigation tool, although the magnitude of the benefit depended on local-scale variations in land use/land cover characteristics.

Related to green infrastructure is blue infrastructure, which directly uses evaporation from water stored and distributed through irrigation, urban water bodies, and low impact development to mitigate heat pollution. Völker et al.[21], Theeuwes et al.[22], and Wu et al.[23] showed that the size, shape, and surrounding land cover of blue infrastructure influences its effectiveness, just like green infrastructure. Indeed, the two have been considered together as blue-green infrastructure solutions[24] for mitigating heat pollution as well as providing a host of other services including sustainable stormwater management[25] and improving the quality of life of the public[26].

A third mitigation strategy, called gray infrastructure, involves the modification of impermeable surfaces (walls, roofs, and pavements) to counter their conventional heating effect[27]. Some examples of gray infrastructure solutions are permeable surfaces, energy-harvesting surfaces, and reflective surfaces. Permeable surfaces store water, which leads to evaporative cooling and thus mitigation of heat pollution[28,29]. Energy-harvesting surfaces, on the other hand, use a series of pipes embedded into surfaces as part of a heat exchange mechanism to directly transfer heat away from those surfaces[30,31] and potentially use it for other applications. Finally, reflective surfaces have a higher albedo than typical construction materials, and hence absorb less solar energy and decrease heat pollution[32]. Among all the possible strategies, the present study focuses on the effectiveness of reflective surfaces as a heat mitigation strategy and therefore, a greater discussion on them is presented below.

Several reflective surface technologies have been proposed to mitigate heat pollution, such as high albedo coatings applied on existing surfaces[33], reflective cementitious materials[34], and reflective roofing tiles[35]. Akbari et al.[36] estimated that a city-wide application of reflective surfaces can lead to a 20% reduction in air conditioning load in the city of Los Angeles. Tewari et al.[37] showed that a city-wide application of reflective roofs in New York and Phoenix can substantially mitigate the combined heat pollution from both the UHI effect and climate change, while Georgescu et al.[38] found similar results in the southwestern US. Santamouris[39] and Qin[40] reviewed a large number of studies and concluded that reflective pavements have a high potential to mitigate heat pollution. Georgescu et al.[41] found that judicious use of reflective surfaces can offset nearly all of the increase in urban air temperatures caused by climate change over the present century.

Several studies have also pointed out that reflective surfaces have limitations and may even be counterproductive in some cases. Sen and Roesler[42] showed that the benefits from them can be highly dependent on the orientation of the neighborhood relative to the prevailing wind direction, while Sen et al.[43] showed that the spatial distribution of reflective surfaces within a neighborhood also affected the extent of mitigation so that not all parts of a neighborhood benefited uniformly from them. Furthermore, depending on their configuration, reflective surfaces can reflect solar radiation onto other surfaces as well as pedestrians, actually leading to an increase in the urban air temperature[44] and cooling energy load[45], and reduced outdoor thermal comfort especially for pedestrians[46–48]. Finally, the albedo of reflective surfaces changes over time due to factors such as deterioration from abrasion, oxidation from exposure to sunlight, and adhesion and accumulation of dirt and debris to the surfaces[49–51]. The period of time during which reflective surfaces are effective may range from a few months to a few years[52], after which it would have to be reapplied.

Despite these limitations, several cities around the world are adopting reflective surfaces as a heat pollution mitigation and climate change adaptation measure, many of them as part of the Global Cool Cities Alliance[53]. For example, in 2018, the city of Los Angeles initiated a project to coat major streets with a commercial reflective coating at a high cost of $40,000 per mile[54]. Their planning relies on studies that are based on large-scale adoption of reflective surfaces, covering entire cities or geographical areas and often neglecting neighborhood-scale effects. This has two serious deficiencies: important local-scale effects may be averaged out when studied over a large scale, and applying reflective surfaces over an entire city or neighborhood would be very difficult if not impossible to achieve given limited resources.

Firstly, as pointed out by Middel et al.[46], many of the shortcomings of reflective surfaces become apparent only when considering local scale complexities in urban neighborhoods. This is often difficult to do, as most analytical and numerical weather models average over these neighborhood-level complexities, and empirical studies can be difficult to perform. A few studies however, have used computational fluid dynamics (CFD) to deterministically model mitigation measures while incorporating neighborhood-level complexity. Sen and Roesler[42] and Georgakis et al.[55] looked at the application of reflective surfaces throughout an idealized urban area, while Sen et al.[43], Dimoudi et al.[56], and Synnefa et al.[57] considered the same in more realistic urban areas. All of these studies used CFD as a prognostic tool to resolve wind speed and air temperature within neighborhoods and found that while reflective surfaces were an effective measure for heat pollution mitigation, their effect varied from one part of the neighborhood to another.

Secondly, both the large-scale and neighborhood-scale studies only considered applying reflective surfaces to all the surfaces in

an entire city or neighborhood to achieve the expected benefits. Due to the high initial as well as maintenance costs, even relatively richer cities have not been able to achieve this in practice, choosing instead to apply reflective surfaces to a few surfaces (such as a few major streets or roofs) that are practically convenient. For a typical city, the goal of modifying 100% of their surfaces may be entirely unfeasible and therefore, they would need to maximize benefits from only a partial application of reflective surfaces. Given that reflective surface-based solutions for mitigation of heat pollution are being applied at the level of individual neighborhoods and even then, mostly limited to only a fraction of total surfaces, guidance is needed to determine an optimal strategy that maximizes benefits with a partial application of reflective surfaces. There is, however, no examination of this problem in the literature, leaving cities and neighborhoods unable to determine how best to make use of limited resources.

In this work, we use a CFD model to examine the feasibility of applying reflective surfaces to only 50% of the area of a prototypical neighborhood with various spatial distributions, for different wind directions and building aspect ratios. We show that there is an optimum distribution that maximizes benefits during the warmest part of the year, which has never been investigated in the literature.

However, we examine the problem purely from the deterministic perspective of reduction in air temperature, without evaluating other effects such as pedestrian thermal comfort, the possibility of combining green, blue, and gray infrastructure solutions, or accounting for statistical uncertainties. Furthermore, the effectiveness of reflective surfaces is examined only with respect to its first application, and subsequent applications and consequent costs (both economic and environmental) to maintain them are not included. These additional aspects will be examined in future work.

## Results

**Cases**. To model the effect of applying reflective surfaces to only a part of the neighborhood, five cases representing different spatial distributions of reflective surfaces were analyzed using the numerical model discussed in Supplementary Note 1, as shown in Fig. 1. In the first case (Conventional case), all the surfaces were assumed to be of existing low albedo materials with a surface temperature $T_{sc}$ (Fig. 1a), while in the second case (Full reflective case), they were assumed to be made of high albedo reflective surfaces with temperature $T_{sr} < T_{sc}$ (Fig. 1b). See the "Methods" section for a discussion of how these temperatures were evaluated. These two cases served as two extreme cases: the first without any reflective surfaces to reduce heat pollution, and the second involving a high investment to make all surfaces reflective.

Three cases with partial application of reflective surfaces were investigated, which represented a smaller investment towards reducing heat pollution but with different spatial distributions. In all these cases, only 50% of the area was set as a reflective surface, with the entire reflective area being concentrated in the upstream part of the neighborhood with respect to a westerly wind (Upstream reflective case), parallel to a westerly wind (Parallel reflective case) or downstream of a westerly wind (Downstream reflective case). These are shown in Fig. 1c–e, respectively.

Each of these five cases of spatial distribution of reflective surfaces was simulated for three building and meteorological configurations:

1. A lower-density building configuration (aspect ratio, $H/W = 1.0$) with wind blowing from west (W)
2. A lower-density building configuration ($H/W = 1.0$) with wind blowing from northwest (NW)
3. A higher-density building configuration ($H/W = 2.0$) with wind blowing from west (W)

Thus, a total of 15 simulations were performed. These were run in parallel on a cluster with 12–16 cores. For each building and meteorological condition, the 2 m air temperatures were extracted (building envelopes were empty, as they were not part of the fluid domain), and the difference between the Conventional case and each of the other four cases (which is called the 2 m air temperature departure of each case) was presented in the form of contour plots for analysis.

*2 m air temperature departures*. For each of the building and meteorological configurations, the 2 m air temperature for the Conventional case is shown in Supplementary Note 4. For the lower-density building configuration ($H/W = 1.0$) with a westerly

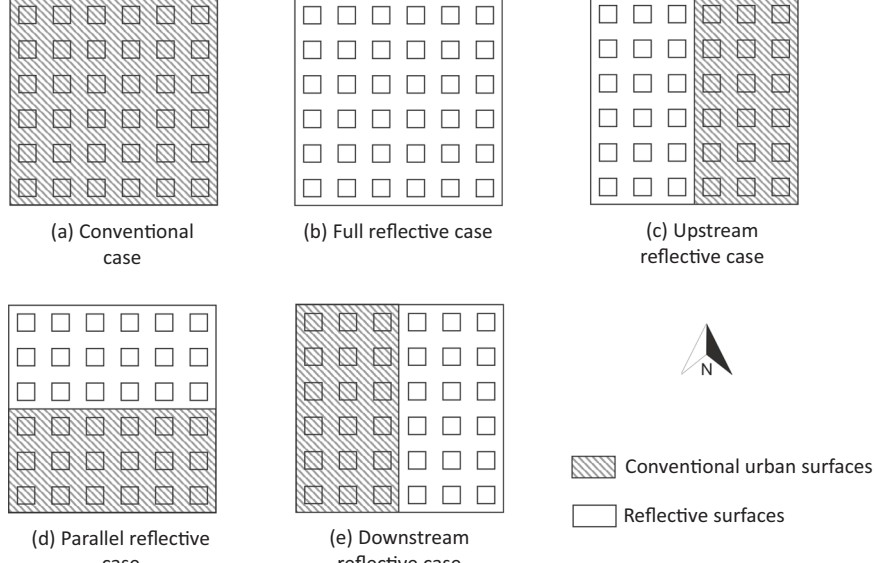

**Fig. 1 Cases analyzed. a** All surfaces being assigned the conventional surface temperature (Conventional case). **b** All surfaces being reflective (Full reflective case). **c** 50% of the surfaces being reflective and directed upstream of a westerly wind (Upstream reflective case). **d** 50% of the surfaces being reflective and directed parallel to a westerly wind (Parallel reflective case) and **e** 50% of the surfaces being reflective and directed downstream of a westerly wind (Downstream reflective case).

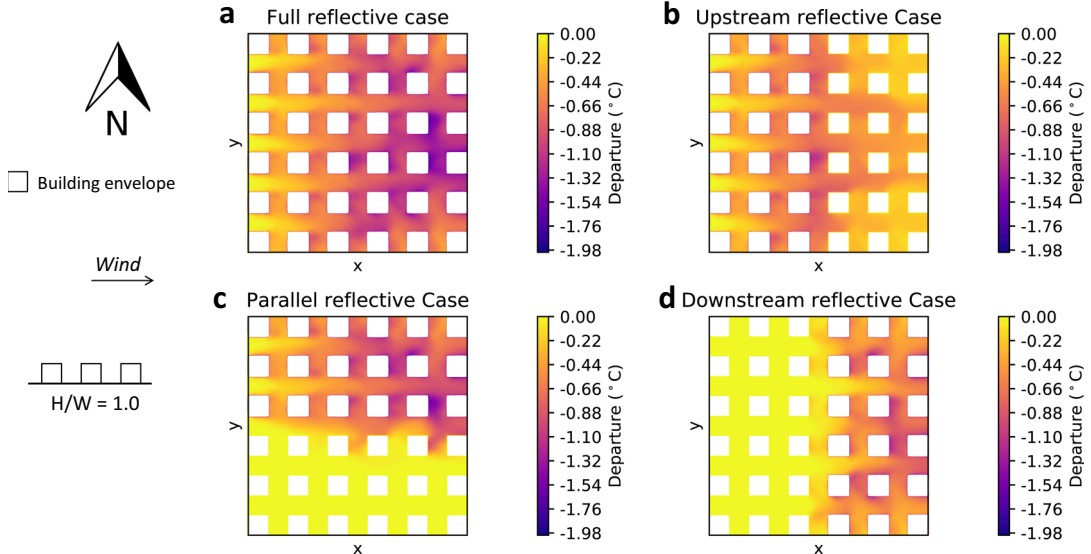

**Fig. 2 2 m air temperature departure from the Conventional case for westerly wind and building aspect ratio H/W = 1.0. a** Full reflective case. **b** Upstream reflective case. **c** Parallel reflective case. **d** Downstream reflective case.

wind, The departure of each case from the Conventional case is shown in Fig. 2. For the Full reflective case (shown in Fig. 2a), the air temperature was reduced throughout the neighborhood, with a maximum decrease of about −1.9 °C, virtually erasing the heat pollution observed in the Conventional case. This shows that large-scale implementation of reflective surfaces is highly effective in reducing heat pollution, although it requires a high investment. For the Upstream, Parallel, and Downstream reflective cases, shown in Fig. 2b–d, respectively, the maximum departure was lower, at about −1.1 °C. However, the departures were distributed differently in space: in the Upstream reflective case, the air temperature was lower not just in the upstream part of the neighborhood but also downstream, as the latter received cooler air. In the Parallel reflective case, the cooling was largely restricted to the part where reflective surfaces were applied, with little benefit in the other part. Finally, in the Downstream reflective case, the cooling was restricted to the downstream area, with the upstream area seeing no benefit.

The departures for the lower-density building configuration (H/W = 1.0) with a northwesterly wind for each case are shown in Fig. 3. For the Full reflective case (Fig. 3a), the maximum departure was about 1.9 °C, the same as the previous configuration. Similarly, for the Upstream, Parallel, and Downstream reflective cases (Fig. 3b–d respectively), the maximum departures were similar to the corresponding cases in the previous configuration, but their spatial distributions were aligned with the new wind direction.

Finally, the departures for the higher-density building configuration (H/W = 2.0) with a westerly wind for each case are shown in Fig. 4. While the maximum magnitude of the departure was similar to the previous two configurations for each case, their spatial distributions were different, with higher departures being concentrated in the wake of buildings on account of relatively lower diffusive mixing.

Additional visualizations of these departures in the form of histograms are shown and discussed in Supplementary Note 5. This is useful to further understand the varying spatial distribution of the cooling effect under the various configurations.

**Benefit-to-cost ratio**. The benefit-to-cost (B/C) ratio, as discussed in the "Methods" section, is an intuitive measure of the

effectiveness of a strategy involving the partial application of reflective surfaces. B/C = 1 indicates that the strategy is as good as the Full reflective case relative to its cost, while B/C > 1 and B/C < 1 indicate that the strategy is better than and worse than the Full reflective case, respectively, relative to its cost. The higher B/C is over 1, the more cost-effective it is for neighborhoods with limited resources to mitigate heat pollution using reflective surfaces.

For the partial application cases corresponding to each wind direction and building aspect ratio configuration, B/C was evaluated and shown in Fig. 5. For all the configurations, B/C > 1 for the Upstream reflective case, with the high-density (H/W = 2.0) westerly wind configuration having the highest value of 1.26, followed by the low-density (H/W = 1.0) westerly wind configuration at 1.18 and the low-density (H/W = 1.0) northwesterly wind configuration at 1.15. In contrast, the B/C for the Parallel reflective case was lower but still ≥1.00, with a value of 1.13 for the low-density (H/W = 1.0) northwesterly wind configuration, 1.03 for the low-density (H/W = 1.0) westerly wind configuration, and 1.00 for the high-density (H/W = 2.0) westerly wind configuration. Finally, the B/C for the Downstream reflective case was consistently below 1.00, with a value of about 0.85 for the low-density (H/W = 1.0) northwesterly wind configuration, 0.82 for the low-density (H/W = 1.0) westerly wind configuration, and 0.74 for the high-density (H/W = 2.0) westerly wind configuration.

From these results, it is apparent that across different configurations of wind direction and building aspect ratio, the spatial distribution of reflective surfaces has a significant effect on their benefit relative to their cost. In the Upstream reflective case, the benefit relative to cost was 15–26% higher than applying reflective surfaces to the entire neighborhood, while it was higher by a smaller 13–0% for the Parallel reflective case. For the Downstream reflective case, the benefit relative to the cost was actually lower by 15–26%. What is striking in these results is that the maximum overall benefit relative to cost was obtained by using reflective surfaces in the upstream part of the neighborhood, whereas the highest heat pollution with conventional surfaces was actually in the downstream part (as shown in Supplementary Note 4). This shows the strong effect of spatial distribution of reflective surfaces on mitigating heat pollution at the neighborhood-scale.

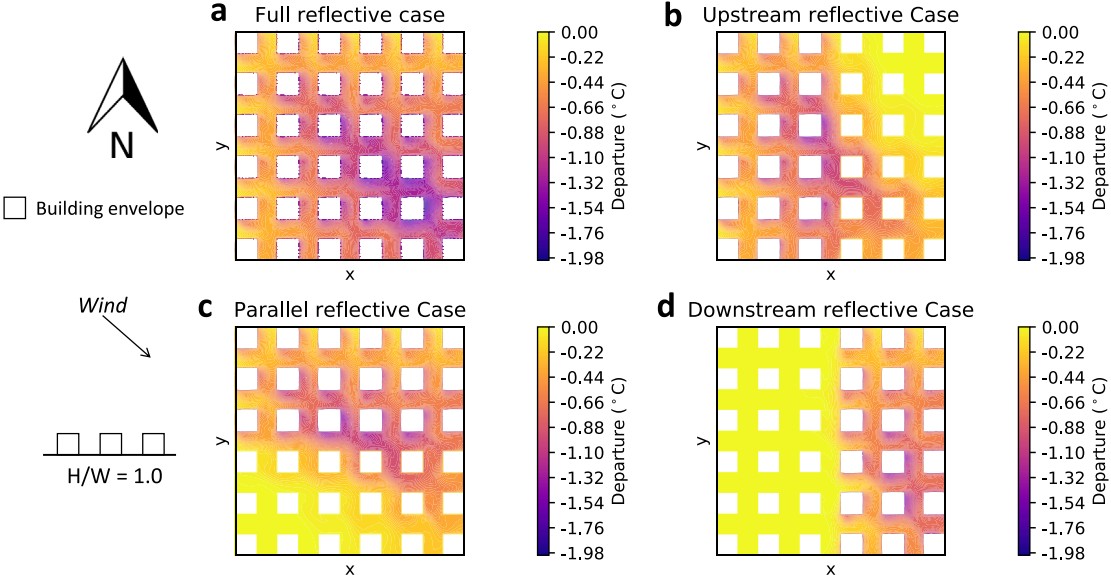

**Fig. 3 2 m air temperature departure from the Conventional case for northwesterly wind and aspect ratio H/W = 1.0. a** Full reflective case, **b** Upstream reflective case. **c** Parallel reflective case. **d** Downstream reflective case.

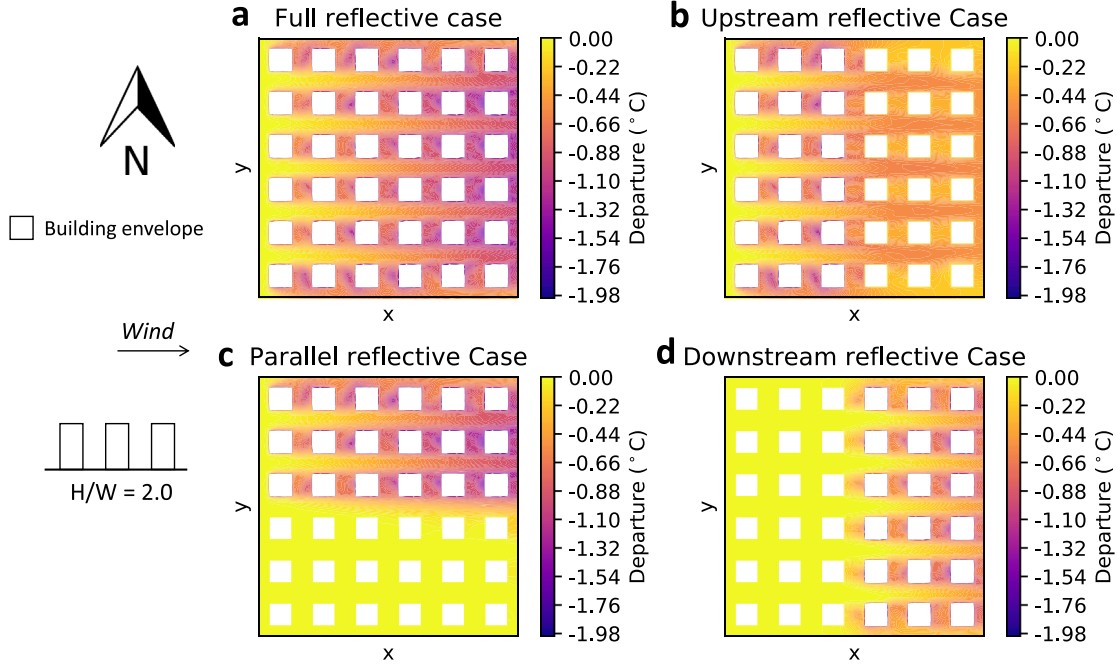

**Fig. 4 2 m air temperature departure from the Conventional case for westerly wind and aspect ratio H/W = 2.0. a** Full reflective case. **b** Upstream reflective case. **c** Parallel reflective case. **d** Downstream reflective case.

## Discussion

From the results, a general principle can be inferred to maximize the benefits from a partial application of reflective surfaces. Reflective surfaces need to be concentrated near the upstream side of the neighborhood in the prevailing wind direction. As a greater proportion of the surfaces are moved towards the downstream, the benefits of reflective surfaces relative to their cost decreases, and this effect is enhanced as the density of the neighborhood increases. Thus, by evaluating the most common wind direction during the time of high heat pollution and applying reflective surfaces accordingly, neighborhoods can maximize heat pollution mitigation even with limited resources.

The present study investigated the effectiveness of reflective surfaces in terms of reduction in 2 m air temperature. Other

aspects such as pedestrian thermal comfort, change in albedo over time, and the environmental impact of manufacturing the reflective surface technologies should also be considered in future work. A combination of solutions involving reflective surfaces, vegetation, and water bodies needs to ultimately be investigated through a specified service life to develop a comprehensive, robust solution to the problem of heat pollution.

## Methods

**Problem setup**. Reflective surfaces were examined for the urban domain described in Supplementary Note 2. The meteorological conditions selected for the analysis represent a high air temperature and low wind speed scenario, where the risk of heat pollution is high and where reflective surfaces would be most beneficial. Such a scenario was described in ref. [42] and reproduced here. The conditions chosen represented the statistically warmest hour of the year in Chicago (July 19 at 3:00 p.m.).

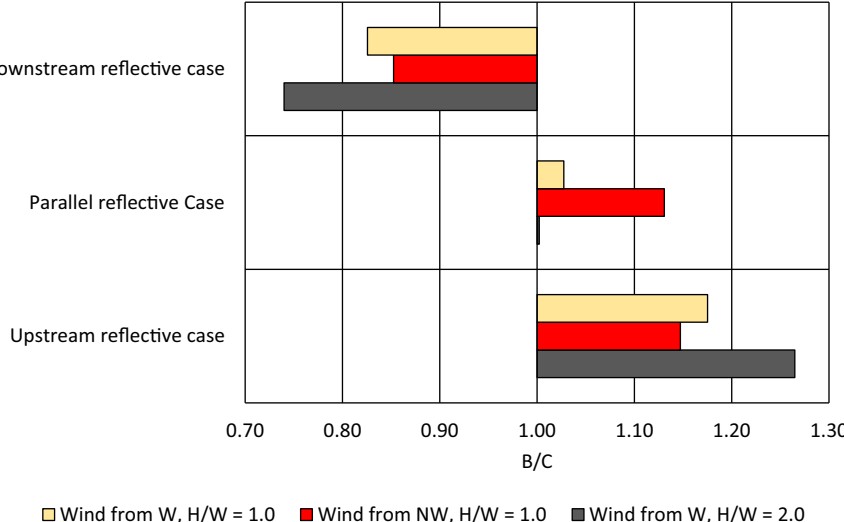

**Fig. 5 Benefit-to-cost ratio B/C for Upstream, Parallel, and Downstream reflective cases.** The B/C ratio is shown for the westerly wind with building aspect ratio of 1.0 in yellow, northwesterly wind with aspect ratio of 1.0 in red, and westerly wind with aspect ratio of 2.0 in grey.

At this hour, the air temperature, which was measured at the airport away from the city's neighborhoods, was $T_a = 35.00\,°C$. Wind was assumed to blow at a speed of $2.0\,m\,s^{-1}$, which was the 20th percentile wind speed in Chicago based on 30 years of observational data from the Illinois State Climatologist Office[58]. Two wind directions were modeled: westerly (W) and north-westerly (NW), with all other directions being symmetric to these two cases.

While these meteorological conditions represent the environment at the boundaries, surfaces inside the neighborhood heat the surrounding air and alter its speed and direction, thus generating heat pollution. The surface temperature depends on the meteorological conditions as well as the albedo of the surface . In a previous study[42], a separate 1D model to resolve surface temperature was developed, and its results were used in the present study. The existing, low albedo (conventional) surfaces, with an albedo of 0.20, were modeled with a surface temperature of $T_{sc} = 41.85\,°C$. Reflective surfaces made of high albedo materials with an albedo of 0.50, were modeled with a surface temperature of $T_{sr} = 35.45\,°C$. The thermal diffusivity and emissivity of these surfaces were assumed to be $0.1\,mm^2\,s^{-1}$ and 0.90, respectively. Although roofs, walls, and roads may have different surface temperatures in practice, they were assumed to be the same in the present study to simplify the analysis. While these values represent one set of possible conditions to assess the effectiveness of reflective surfaces, the same approach can easily be used to assess any other set of conditions that are more representative of other neighborhoods, cities, and urban surfaces.

When these conditions are implemented in the numerical model as boundary conditions, the 3D wind speed and air temperature fields are obtained, which need to be further processed to examine the extent of heat pollution. This is generally quantified based on the air temperature at 2 m height, called the canopy height, where the most human outdoor activity takes place[59]. Therefore, after each simulation, the 2 m air temperature was extracted from the full-field solution. From here, the departure of each case from the Conventional case (no reflective surfaces) was evaluated for further analysis.

**Benefit-to-cost ratio**. Observing the distribution of departures is useful to qualitatively explain spatial differences in heat pollution. However, a more quantitative explanation allows for a direct comparison between the different cases and configurations. Two principles underlay this quantitative approach. First, the total benefit derived from any reflective surface strategy depends on both the magnitude of the decrease in 2 m air temperature as well as the area over which that decrease is spread. Second, the Full reflective case represents the maximum benefit possible for the neighborhood and reflective surfaces in this study, albeit at a high cost.

Thus, the total benefit of any case can be evaluated as the area integral of the departures shown in Figs. 2–4, while the relative benefit $B$ of any case relative to the Full reflective case is given by the ratio of the total benefit of the two cases. Mathematically, if $T_c$, $T_r$, and $T$ represent the 2 m air temperature field of the Conventional case, the Full reflective case, and any one of partial cases, respectively for the same wind direction and building aspect ratio, then $B$ is given by Eq. (1). Importantly, area $A$ over which the departures are integrated was that of the urban neighborhood alone, and not the extended numerical domain around it, since the aim is to maximize benefits within the neighborhood. $B$ for the Conventional case, in which no mitigation measures were applied, was 0, while that for the Full

reflective case, where the maximum mitigation was applied, was 1.

$$B = \frac{\int_A (T_c - T)\mathrm{d}A}{\int_A (T_c - T_r)\mathrm{d}A} \tag{1}$$

The relative cost $C$ of a case can be expressed in terms of the surface area of the neighborhood that has to be modified into a reflective surface relative to the total area, for the same aspect ratio. For the Full reflective case, $C$ was 1, and for the Upstream, Parallel, and Downstream reflective cases, it was 0.5, as only half the total area was modified. Having calculated $B$ and $C$ for each case, the benefit-to-cost ratio ($B/C$) could be evaluated.

## Data availability
The datasets generated and/or analyzed during the current study are available from the corresponding author on reasonable request. No external datasets were analyzed for this study.

## Code availability
The code for generating the mesh as well as the OpenFOAM case files are available from https://github.com/sushobhansen/partial-cool-surfaces. The repository contains instructions for running the code.

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

## Acknowledgements

Postdoctoral funding to support this study was provided by the Anthony Gill Endowed Chair at the University of Pittsburgh. The authors are grateful to the Center for Research Computing at the University of Pittsburgh for computing resources.

## Author contributions

S.S. contributed to conception and design of the work, acquisition and analysis of data, interpretation of data, and drafting and revision of the manuscript; L.K. contributed to conception of the work, interpretation of data, and revision of the manuscript.

## Competing interests

The authors declare no competing interests.
