## [Peer Review File · Nature Communications]

REVIEWER COMMENTS

Reviewer #1 (Remarks to the Author):

- Timely topic. Though the study provided some information on the utility of the analytical process, the scope was very limited to deliver significant contribution.
- Standard English technical writing is mandated. There are a few words that are used in the wrong context. For example, page 2, paragraph 2, line 7: the word exacerbate is not intertwined into the sentence. Similarly, many others.
- Are cool surfaces always reflective? What about porous asphalt? They are fundamentally blacktopping materials, but are cooler. Comment.
- Figures 4 and 5: how significant is each degree Celsius? How does one map the significance / sensitivity in real case scenario? Also, there should be some distinguishing colors used to characterize what is being assumed.
- Conclusion: what do the authors mean by "climate change is predicted to exacerbate heat pollution..."? There could be communities where climate change is actually an advantage. Comment. Again, are all cool surfaces reflective?
- What is climate justice? Explain.
- What is novel about the study? A thorough discussion is required regarding what the authors accomplished and what is already available in literature.
- An interlink between the stated hypothesis and findings is missing. Are authors targeting a particular community since the title seems hazy.
- Please state the need for using CFD for analysis? Are there other methodologies available in the literature to undertake similar analysis? How confident are the authors with the proposed model? What about statistical measures of accuracy?
- What is the significance of this research study? A separate section is required.

Reviewer #2 (Remarks to the Author):

I am concerned about the strong statement authors make about this study being an important step in improving living conditions in disadvantaged neighborhoods. If this study was about improving the conditions of disadvantaged neighborhoods, I would have expected a broader analysis on the impacts (pros and cons) of various strategies to mitigate UHI. I agree with the authors that the issue of UHI is an issue of climate equity and justice however the authors could be more objective on the proposed solution. Decisionmakers need to understand how to effectively use the various strategies which this study helps to advance, however equally important is to understand the tradeoffs that exist and effectiveness in comparison to other strategies (which are a major limitation of this study and limitation to making a strong connection to improving living conditions of disadvantaged neighborhoods). Authors mention that there are other strategies however they do not justify why they are only modeling one. One particular strategy is increasing the amount of vegetative surfaces. In urban planning there is a shift to revitalizing the urban environment to be more attractive and inviting to pedestrians and habitants by increasing the green space. This strategy has multiple benefits such as potential to increase active transportation, happiness, as well as reducing UHI (see <https://www.sciencedirect.com/science/article/abs/pii/S0013935117315876> on benefits of green space and human perception of temperature based on ego depletion). With the emphasis on climate justice, I found it odd how this strategy was not evaluated due to the many additional benefits that come with green space. Regardless, decisionmakers need tools to evaluate all of the various UHI mitigation strategies to assess which is the most appropriate for their situation. I can see how this study is a step towards that, however I do not think that the authors articulate the complexity of the problem well. Authors do not comment on any of the well known tradeoffs that exist with reflective coatings (especially for pavements-see <https://iopscience.iop.org/article/10.1088/1748-9326/ab87d4>) and only evaluate the gains. To truly improve the living conditions of the people in these urban areas, the potential tradeoffs must also be evaluated. That being said I believe that the authors have made a contribution to the body of knowledge in their advancements of modeling. I strongly urge the authors to take an more objective approach with the focus of the paper on the advancements they made in the model development rather than making some of the broader statements linking this to climate justice since this was really not studied. Furthermore, authors should acknowledge the potential tradeoffs that exist and clearly articulate the limitations of this study in assessing these tradeoffs and any model assumptions. Lastly, it would be good for authors to provide further insights on what are potential next steps to further advance this technique and address any limitations noted advancing towards a more wholistic evaluation with pros and cons, as well as incorporation of other mitigation strategies.

Minor comment:

What is considered "high albedo" and "low albedo" for the purposes of this study. This would help readers understand the magnitude of the proposed changes.

Reviewer #3 (Remarks to the Author):

The authors have provided further evidence that cool surfaces can lower the UHI and a partial application of these surfaces can be very effective if strategically located with the prevailing wind direction. The paper is organized and written well. This paper can be very useful to policymakers, planners, and engineers who need to determine rational approaches to allocate cool surfaces to cities that have heat pollution. This paper's introduction of partial area-wide surface coating and B/C analysis is a simple but effective approach to communicate a very realistic strategy with constrained resources.

The data and approach are valid for this urban form. How would the heights of the buildings affect the conclusions made in the paper regarding upstream coatings, e.g., if $H/W = 5$ to 10 what B/C would be realized? What would be the conclusion if the authors implemented the analysis for all wind directions and left the upstream position of the reflective surface in the position defined in this paper? Would the B/C > 1 or something less than 1.0 given 80% of the wind does not come from the direction used in the analysis. Likewise how would this affect the temperatures potential in the winter, i.e., would these homes unintentionally have higher heating bills? Authors may need to narrow their conclusions to certain assumptions to avoid too broad of a conclusion in all urban forms and seasons.

Authors should list the the values of high and low albedo and define the thermal diffusivities and emissivities used in the analysis.

The temperature departure figures are nice to visualize trends spatially. However, would it also be helpful to see all the data in one figure by plotting the temperature departure for a given control volume versus number of occurrences. The color contrast in figure 5 make it hard to determine if most control volumes depart a little bit or a lot e.g., $> 1C$ or $1.5C$. A plot of distribution of temperature departures vs. number of times occurring would give some sense for the different cases how effective cool strategies are for certain volume size. This gives a plot that can be compared against cases and the distribution of the departures within a case to other cases. This may be similar to the benefit calculation and possibly not as informative as a single number like B.

Reviewer #1:

Comment: Timely topic. Though the study provided some information on the utility of the analytical process, the scope was very limited to deliver significant contribution.

Response: Thank you for the comment. We have significantly expanded the results presented and have more clearly shown how it fits into the existing state of knowledge.

Comment: Standard English technical writing is mandated. There are a few words that are used in the wrong context. For example, page 2, paragraph 2, line 7: the word exacerbate is not intertwined into the sentence. Similarly, many others.

Response: Thank you for pointing this out. The language used throughout the manuscript has been checked and revised so that it is more specific and used in the correct context. The specific line cite by the reviewer has been modified as follows:

Furthermore, studies have predicted that climate change will intensify the negative effects of heat pollution in the future in some regions by increasing both the intensity and frequency of extreme heat events [21-23], although it could lead to some benefits such as a reduction in cold-related deaths in other regions [24].

Similar changes have been incorporated in the rest of the manuscript.

Comment: Are cool surfaces always reflective? What about porous asphalt? They are fundamentally blacktopping materials, but are cooler. Comment.

Response: This is a good point, in part it reflects nomenclature that, while used extensively, may lack specificity, and in part that there are a variety of heat pollution mitigation solutions. Based on this comment as well as that from Reviewer #2, the manuscript now goes into more details about mitigation solutions, broadly classified as green, blue, and grey infrastructure. In particular, in response to this comment, the manuscript makes the following observation:

A third mitigation strategy, called grey infrastructure, involves the modification of impermeable surfaces (walls, roofs, and pavements) to counter their conventional heating effect [34]. Some examples of grey infrastructure solutions are permeable surfaces, energy-harvesting surfaces, and cool surfaces. Permeable surfaces store water, which leads to evaporative cooling and thus mitigation of heat pollution [35, 36]. Energy-harvesting surfaces, on the other hand, use a series of pipes embedded into surfaces as part of a heat exchange mechanism to directly transfer heat away from those surfaces [37, 38] and potentially use it for other applications. Finally, cool surfaces, also called reflective surfaces, have a higher albedo than typical construction materials, and hence absorb less solar energy and decrease heat pollution [39].

Thus, while all grey infrastructure solutions aim to reduce air and surface temperature (hence they are “cool”), cool surfaces have come to be synonymous with reflective surfaces in much of the literature and the two terms are used interchangeably in this paper.

Comment: Figures 4 and 5: how significant is each degree Celsius? How does one map the significance / sensitivity in real case scenario? Also, there should be some distinguishing colors used to characterize what is being assumed.

Response: Thank you for the comment. The following illustration of the significance of elevated air temperatures has been added:

Previous studies have shown that for every 1°C increase in air temperature, residential water consumption for a family can increase by about 2,000 liters per month [9], monthly electricity consumption in the hottest months can increase by up to 8.5% [8], and the potential for thermal discomfort and heat-related morbidity can also increase significantly [14]. Thus, heat pollution to the extent of 2.5°C can be quite significant. For disadvantaged communities lacking in air conditioning and other mitigation measures, this temperature may lead to serious heat-related consequences.

As to the colors in the figures, the author have changed the color scheme so that warmer temperatures are in shades of red and cooler temperatures in shades of blue, which should make it easier to follow.

Comment: Conclusion: what do the authors mean by “climate change is predicted to exacerbate heat pollution...”? There could be communities where climate change is actually an advantage. Comment. Again, are all cool surfaces reflective?

Response: Thank you for the comment, it is a valid observation. In this paper, we focused on situations where climate change would have negative consequences in the form of elevated air temperatures, although it is true that in some places, higher (but not elevated or too high) air temperatures can be beneficial. The manuscript has been modified to be more specific:

Furthermore, studies have predicted that climate change will intensify the negative effects of heat pollution in the future in some regions by increasing both the intensity and frequency of extreme heat events [21-23], although it could lead to some benefits such as a reduction in cold-related deaths in other regions [24].

Cool and reflective surfaces are largely used synonymously in the literature as explained in response to a previous comment. We followed that convention in this manuscript.

Comment: What is climate justice? Explain.

Response: Based on Reviewer #2's comments, we have removed this reference.

Comment: What is novel about the study? A thorough discussion is required regarding what the authors accomplished and what is already available in literature.

Response: Thank you for the comment. The authors have included a separate section about the significance of the study:

Given that cool surface-based solutions for mitigation of heat pollution are being applied at the level of individual neighborhoods and even then, disadvantaged neighborhoods would only be able to apply them to a part of their surfaces, guidance is needed to determine an optimal strategy that maximizes benefits with a partial application of

cool surfaces. There is however, no examination of this problem in the literature, leaving cities and neighborhoods unable to determine how best to make use of limited resources.

The present study uses a CFD model to examine the feasibility of applying cool surfaces to only a portion of the area of a prototypical neighborhood for different wind direction and building aspect ratio configurations. In particular, this approach was used to study the effect of applying cool surfaces to 50% of the neighborhood with different spatial distributions to find an optimum distribution that maximizes benefits, which has never been investigated in the literature.

In short, the problem of limited capacity to apply cool surfaces exists right now, and while literature has discussed the benefits of applying cool surfaces, there has been no study on optimal application strategies especially down at the neighborhood scale. The present study fills this gap. As summarized in the conclusion, we have derived an optimal strategy and provided a general principle that cities can use to maximize benefits from cool surfaces.

Comment: An interlink between the stated hypothesis and findings is missing. Are authors targeting a particular community since the title seems hazy.

Response: Apropos to the response to the previous comment where the goals of the study were stated, the conclusion makes the following observations:

While downstream areas experienced the highest heat pollution for different wind directions and aspect ratios, applying cool surfaces to only that part was less effective relative to cost than applying cool surfaces to the entire area. On the other hand, placing more of the cool surfaces in the upstream part of the neighborhood yielded disproportionately higher benefits relative to cost, as upstream cooling led to cooler air blowing into the downstream part of the neighborhood as well, spreading out the benefits over a larger area. From these results, it may be concluded that neighborhoods can maximize the benefit of a limited application of cool surfaces by concentrating them as close as possible to the upstream part of the neighborhood in the most common direction of wind in the warmest months.

Thus, while the goal of the study was to develop an optimal strategy for applying cool surfaces to 50% of a neighborhood, the findings concluded that the optimal strategy was to concentrate the cool surfaces to the upstream part of the neighborhood. Thus, we have tied the goals and findings together. The study was aimed at a general, prototypical urban neighborhood (see response to previous comment) to develop a general strategy. Specific strategies tailored to individual cities will be investigated in the future.

Comment: Please state the need for using CFD for analysis? Are there other methodologies available in the literature to undertake similar analysis? How confident are the authors with the proposed model? What about statistical measures of accuracy?

Response: The manuscript has been updated to state the benefits of using CFD as follows:

Firstly, as pointed out by [53], many of the shortcomings of cool surfaces become apparent only when considering local scale complexities in urban neighborhoods. This is often difficult to do, as most analytical and numerical weather models average over these neighborhood-level complexities, and empirical studies can be difficult to perform. A few studies however, have used Computational Fluid Dynamics (CFD) to deterministically model

mitigation measures while incorporating neighborhood-level complexity. Sen & Roesler [49] and Georgakis et al. [57] looked at the application of cool surfaces throughout an idealized urban area, while Sen et al. [50], Dimoudi et al. [58], and Synnefa et al. [59] considered the same in more realistic urban areas. All of these studies used CFD as a prognostic tool to resolve wind speed and air temperature within neighborhoods and found that while cool surfaces were an effective measure for heat pollution mitigation, their effect varied from one part of the neighborhood to another.

The advantage of using CFD is that it does not average over local-scale effects as is the case with many analytical and large-scale weather forecasting tools and is easier to use than empirical studies for practical reasons. Furthermore, CFD is a prognostic method that does not rely on statistical correlations, instead directly solving physical equations, and thus causation can be established. That said, the manuscript describes the limitation of using CFD as follows:

Furthermore, while CFD allows for resolving local-scale complexities, it is a deterministic method that does not account for uncertainties in the solution. This can be overcome through approaches such as ensemble modeling however, this was outside the scope of the present study.

Being deterministic, statistical measures of accuracy are not applicable to CFD analyses. Instead, accuracy is demonstrated through convergence and validation against hydrodynamically similar data obtained from experiments. For this reason, the manuscript includes details on convergence and validation of the model, both of which demonstrate that the model is reasonably accurate. This is the approach used by other studies that use CFD for modeling the urban climate as well and thus, we are reasonably confident that the approach is valid, and the results are accurate.

Comment: What is the significance of this research study? A separate section is required.

Response: This comment has been addressed in previous comments above regarding novelty, stated hypothesis, and findings. In short, the problem of limited application of cool surfaces by cities already exists (and is inevitable for socio-economically disadvantaged neighborhoods that lack resources for large-scale implementation), and this study shows that there is an optimal way to maximize the benefits. The study is thus significant as it informs and improves a heat mitigation approach that is already underway.

Reviewer #2:

Comment: I am concerned about the strong statement authors make about this study being an important step in improving living conditions in disadvantaged neighborhoods. If this study was about improving the conditions of disadvantaged neighborhoods, I would have expected a broader analysis on the impacts (pros and cons) of various strategies to mitigate UHI. I agree with the authors that the issue of UHI is an issue of climate equity and justice however the authors could be more objective on the proposed solution. Decisionmakers need to understand how to effectively use the various strategies which this study helps to advance, however equally important is to understand the tradeoffs that exist and effectiveness in comparison to other strategies (which are a major limitation of this study and limitation to making a strong connection to improving living conditions of disadvantaged neighborhoods). Authors mention that there are other strategies however they do not justify why they are only modeling one. One particular strategy is increasing the amount of vegetative surfaces. In urban planning

there is a shift to revitalizing the urban environment to be more attractive and inviting to pedestrians and habitants by increasing the green space. This strategy has multiple benefits such as potential to increase active transportation, happiness, as well as reducing UHI (see <https://www.sciencedirect.com/science/article/abs/pii/S0013935117315876> on benefits of green space and human perception of temperature based on ego depletion). With the emphasis on climate justice, I found it odd how this strategy was not evaluated due to the many additional benefits that come with green space.

Response: Thank you for the excellent comment. The reviewer is correct in that there are several other strategies aside from cool surfaces, the benefits of which go beyond only reducing air temperature. We have thoroughly revised the discussion, explaining different heat mitigation approaches (green, blue, and grey infrastructure) to place the present study in its proper context. We have also limited the discussion on climate equity and justice to make the limited point that socio-economically disadvantaged neighborhoods typically suffer the most from elevated temperatures but have the least resources for mitigation. We believe that this point is important to demonstrate the need to balance both benefits and costs of various solutions, and the approach used in this study for cool surfaces is a step in that direction. We have added the following observation to explicitly state this limited scope of our work:

A limitation of this study is that it examines the problem purely from the perspective of reduction in air temperature, without evaluating other effects such as pedestrian thermal comfort, or the possibility of combining green, blue, and grey infrastructure solutions for a more comprehensive solution. These additional aspects will be examined in future work.

Furthermore, we have also modified the title of the manuscript to reflect this more limited scope.

Comment: Regardless, decisionmakers need tools to evaluate all of the various UHI mitigation strategies to assess which is the most appropriate for their situation. I can see how this study is a step towards that, however I do not think that the authors articulate the complexity of the problem well. Authors do not comment on any of the well known tradeoffs that exist with reflective coatings (especially for pavements-see <https://iopscience.iop.org/article/10.1088/1748-9326/ab87d4>) and only evaluate the gains. To truly improve the living conditions of the people in these urban areas, the potential tradeoffs must also be evaluated. That being said I believe that the authors have made a contribution to the body of knowledge in their advancements of modeling. I strongly urge the authors to take an more objective approach with the focus of the paper on the advancements they made in the model development rather than making some of the broader statements linking this to climate justice since this was really not studied. Furthermore, authors should acknowledge the potential tradeoffs that exist and clearly articulate the limitations of this study in assessing these tradeoffs and any model assumptions.

Response: Thank you again for the excellent and insightful comment. It is indeed true that studies on heat mitigation solutions tend to use a narrow metric without considering other, possibly counterproductive effects, and yet it is important to advance the ability to use even this narrow metric to provide some guidance at a time when cities are actively adopting such solutions. As mentioned in the response to the previous comment above, the manuscript now acknowledges this shortcoming, and

limits the discussion on climate equity and justice to demonstrate the need for neighborhood-level strategies that consider the limited capacity of disadvantaged neighborhoods. We have also highlighted some studies that demonstrate the inherent complexity of the problem at hand and its various tradeoffs:

Several studies have pointed out that cool surfaces have limitations and may even be counterproductive in some cases. Sen and Roesler [49] showed that the benefits from them can be highly dependent on the orientation of the neighborhood relative to the prevailing wind direction, while Sen et al. [50] showed that the spatial distribution of cool surfaces within a neighborhood also affected the extent of mitigation so that not all parts of a neighborhood benefited uniformly from them. Furthermore, depending on their configuration, cool surfaces can reflect solar radiation on to other surfaces as well as pedestrians, actually leading to an increase in the urban air temperature [51] and cooling energy load [52], and reduced outdoor thermal comfort especially for pedestrians [53-55].

We hope that with these changes, the manuscript has achieved a balance in terms of complexity of the problem and development of some general guidelines that address one aspect of it. This is all the more important since many cities are already adopting cool surfaces for heat mitigation and climate change adaptation, and these guidelines would be of immediate value to them.

Comment: Lastly, it would be good for authors to provide further insights on what are potential next steps to further advance this technique and address any limitations noted advancing towards a more wholistic evaluation with pros and cons, as well as incorporation of other mitigation strategies.

Response: In addition to the response to the comments above, the authors have added the following observation to the very end of the manuscript in response to this comment:

The present study investigated the effectiveness of cool surfaces in terms of reduction in 2 m air temperature. Other aspects such as pedestrian thermal comfort should also be considered in future work. A combination of solutions involving cool surfaces, vegetation, and water bodies needs to ultimately be investigated to develop a comprehensive, robust solution to the problem of heat pollution.

Comment: What is considered "high albedo" and "low albedo" for the purposes of this study. This would help readers understand the magnitude of the proposed changes.

Response: In this study, low albedo was 0.20, which is the typical albedo of (somewhat aged) construction materials, while high albedo was 0.50, corresponding to the typical albedo of reflective coatings/cementitious materials reported in the literature and commercial products. The manuscript has been updated with this information:

The existing, low-albedo (conventional) surfaces, with an albedo of 0.20, were modeled with a surface temperature of $T_{sc} = 41.85^{\circ}\text{C}$. Cool surfaces made of reflective (high albedo) materials, with an albedo of 0.50, were modeled with a surface temperature of $T_{sr} = 35.45^{\circ}\text{C}$. The thermal diffusivity and emissivity of these surfaces was assumed to be $0.1 \text{ mm}^2 \text{ s}^{-1}$ and 0.90 respectively.

Reviewer #3:

Comment: The authors have provided further evidence that cool surfaces can lower the UHI and a partial application of these surfaces can be very effective if strategically located with the prevailing wind direction. The paper is organized and written well. This paper can be very useful to policymakers, planners, and engineers who need to determine rational approaches to allocate cool surfaces to cities that have heat pollution. This paper's introduction of partial area-wide surface coating and B/C analysis is a simple but effective approach to communicate a very realistic strategy with constrained resources.

Response: Thank you for your comments.

Comment: The data and approach are valid for this urban form. How would the heights of the buildings affect the conclusions made in the paper regarding upstream coatings, e.g., if $H/W = 5$ to 10 what B/C would be realized? What would be the conclusion if the authors implemented the analysis for all wind directions and left the upstream position of the reflective surface in the position defined in this paper? Would the $B/C > 1$ or something less than 1.0 given 80% of the wind does not come from the direction used in the analysis.

Response: Thank you for the comment. We have performed additional simulations, with a different wind direction (northwest, note that with westerly and northwesterly winds, any other wind direction can be ascertained from symmetry) and $H/W = 2$ (higher values of H/W are somewhat difficult to perform as the computational costs become restrictive, but the authors have explained the physical effect of increasing the aspect ratio and these results can be extrapolated to higher H/W based on that). After looking at all the data, we conclude that $B/C > 1$ for upstream cases across the different wind and building configurations, and $B/C < 1$ for downstream cases. Thus, the general principle derived in this paper (i.e., cool surfaces should be concentrated in the upstream) still stands after the additional analysis.

Comment: Likewise how would this affect the temperatures potential in the winter, i.e., would these homes unintentionally have higher heating bills? Authors may need to narrow their conclusions to certain assumptions to avoid too broad of a conclusion in all urban forms and seasons.

Response: Thank you for the comment. The scope of the present study is restricted to the warmest months of the year when heat pollution becomes a significant problem:

The present study uses a CFD model to examine the feasibility of applying cool surfaces to only a portion of the area of a prototypical neighborhood for different wind direction and building aspect ratio configurations. In particular, this approach was used to study the effect of applying cool surfaces to 50% of the neighborhood with different spatial distributions to find an optimum distribution that maximizes benefits during the warmest part of the year, which has never been investigated in the literature.

The reviewer brings up an important question of the corresponding effect on heating loads in the winter. While this is out of scope of the present work, it has been examined by other authors e.g., Hosseini & Akbari (Energy and Buildings 2016) and Baral et al. (J of Cleaner Production 2018), have shown that the gains from reduced cooling loads are typically higher than penalties from increased heating loads even in colder climates. Of course, this depends on local-scale effects as well. Since the

present study does not investigate this aspect, we have ensured that the conclusions reflect that the results are only for the warmest months of the year.

Comment: Authors should list the the values of high and low albedo and define the thermal diffusivities and emissivities used in the analysis.

Response: The manuscript has been modified to include this information:

The existing, low-albedo (conventional) surfaces, with an albedo of 0.20, were modeled with a surface temperature of $T_{sc} = 41.85^{\circ}\text{C}$. Cool surfaces made of reflective (high albedo) materials, with an albedo of 0.50, were modeled with a surface temperature of $T_{sr} = 35.45^{\circ}\text{C}$. The thermal diffusivity and emissivity of these surfaces was assumed to be $0.1 \text{ mm}^2 \text{ s}^{-1}$ and 0.90 respectively.

Comment: The temperature departure figures are nice to visualize trends spatially. However, would it also be helpful to see all the data in one figure by plotting the temperature departure for a given control volume versus number of occurrences. The color contrast in figure 5 make it hard to determine if most control volumes depart a little bit or a lot e.g., $>1\text{C}$ or 1.5C . A plot of distribution of temperature departures vs. number of times occurring would give some sense for the different cases how effective cool strategies are for certain volume size. This gives a plot that can be compared against cases and the distribution of the departures within a case to other cases. This may be similar to the benefit calculation and possibly not as informative as a single number like B.

Response: Thank you for the excellent suggestion. The authors have added temperature departure histograms (weighted by area of the control surface at 2 m height) for all the cases and configurations studied. As the reviewer will see in the manuscript, this does provide additional insight into the spatial differences between the different strategies, leading up to the calculation of the benefit B. It thus makes for a good bridge between the raw departures and the integrated value.

REVIEWER COMMENTS

Reviewer #2 (Remarks to the Author):

I appreciate the changes made to address the comments. One concern that I have which still remains is the fact that the albedo does change over time and can be costly to maintain (economically and environmentally). I think it would be worthwhile to note how uncertainty analysis or a life cycle approach could be part of future work. In addition, I think this helps further promote the strategic application approach demonstrated.

Referee #1 could not look at the manuscript again. Instead, I (the editor) asked referee #2 to look at your revisions towards the comments of referee #1. Referee #2 states that you should tone down the part about disadvantaged neighbourhoods even more - I copied the comments of referee #2 below - please answer to them as well:

I have reviewed the comments from reviewer one. I think one of the issues with the paper that I know I picked up on and this reviewer 1 seems to have brought up some element is the tie to disadvantaged neighborhoods. Yet this is true that disadvantaged neighborhoods may be having more of the hardship from climate change, I don't think this should be so much of the focus of the study since I believe it takes away from the contribution of the study by mudding the study's focus. I know that the authors did take out some of this emphasis by removing this from the title but in the response to one of reviewer 1 comments the authors mention the following: "Given that cool surface-based solutions for mitigation of heat pollution are being applied at the level of individual neighborhoods and even then, disadvantaged neighborhoods would only be able to apply them to a part of their surfaces, guidance is needed to determine an optimal strategy that maximizes benefits with a partial application of cool surfaces. There is however, no examination of this problem in the literature, leaving cities and neighborhoods unable to determine how best to make use of limited resources."

The issue of determining an optimal strategy that maximizes the benefits with a partial application is applicable whether it is a disadvantage neighborhood or not. I don't think adding the disadvantaged neighborhood in this paragraph for motivation is needed.

With regards to the comment of "Are cool surfaces always reflective?" The authors added the noted paragraph which is a good improvement to address the comment, however I think they should be clearer and not use "cool surfaces" interchangeably with "reflective pavements". I think this is misleading since there could be other cool surfaces that are not reflective.

Reviewer #3 (Remarks to the Author):

The author have improved the quality of their argument and claims with their revisions including a very thorough and referenced response to reviewer comments. The introduction has been improved to clarify main contributions in the grey infrastructure area vs. green-blue. The reviewer has not reason not to recommend for publication. A few general comments that the author could address are the following.

For figure 5a and 5b it seems like the colors show that the temperature departure at the building inlet is increase the 2m air temperature relative to Figure 4, which is the control. For 5a and 5b, is the 2m air temperature departure on western buildings <0 or >0. If the 2m temp is greater than zero then there seems to be something wrong with model given the cool surfaces that are placed 100% and upstream respectively for these 2 cases. The red and orange color difference in Figure 5 make it hard to know if it is positive or negative temperature departure. For Figure 5c and 5d the temperature departure away from the cool surface, the temperature departure should be greater than zero.

The reviewer likes Figure 6 (and other similar figures) as an improved way of communicating areas seeing temperature departures relative to cool surfaces. The additional histograms and enhancements to the visual improve the understanding of potential to increase the benefit of cool surfaces

Overall well written and described paper that can assist policymakers, planners, and engineers target cool surface strategies beyond current thinking which is 100% coverage recommendation by product manufacturers.

The B/C also helps to see the upstream placement of the surfaces maximizes benefit in the targeted neighborhood.

Authors could add some numbers in conclusions particularly about benefit/cost < 1.0 to cool surfaces not strategically placed.

Response to reviewers

Reviewer #2:

Comment: I appreciate the changes made to address the comments. One concern that I have which still remains is the fact that the albedo does change over time and can be costly to maintain (economically and environmentally). I think it would be worthwhile to note how uncertainty analysis or a life cycle approach could be part of future work. In addition, I think this helps further promote the strategic application approach demonstrated.

Response: Thank you for pointing this out. We have modified the manuscript to address this comment as follows. The decrease in albedo of reflective surfaces over time has been added as follows:

Finally, the albedo of reflective surfaces changes over time due to factors such as deterioration from abrasion, oxidation from exposure to sunlight, and adhesion and accumulation of dirt and debris to the surfaces [49-51]. The period of time during which reflective surfaces are effective may range from a few months to a few years [52], after which it would have to be reapplied.

The shortcoming of the present study with respect to this was added to its limitations:

Finally, the effectiveness of reflective surfaces is examined only with respect to its first application, and subsequent applications and consequent costs (both economic and environmental) to maintain them are not included. This could be studied in future work through a life cycle analysis.

Finally, future work in the conclusion was modified as follows:

The present study investigated the effectiveness of reflective surfaces in terms of reduction in 2 m air temperature. Other aspects such as pedestrian thermal comfort, change in albedo over time, and the environmental impact of manufacturing the reflective surface technologies should also be considered in future work. A combination of solutions involving reflective surfaces, vegetation, and water bodies needs to ultimately be investigated through a specified service life to develop a comprehensive, robust solution to the problem of heat pollution.

We agree that the benefits of strategically applying reflective surfaces to only a part of the area would yield additional benefits in a life cycle assessment (for example, through decreased emissions from manufacturing coatings), and would serve as an interesting study in the future.

Reviewer #2's comments on the previous response to Reviewer #1:

Comment: I have reviewed the comments from reviewer one. I think one of the issues with the paper that I know I picked up on and this reviewer 1 seems to have brought up some element is the tie to disadvantaged neighborhoods. Yet this is true that disadvantaged neighborhoods may be having more of the hardship from climate change, I don't think this should be so much of the focus of the study since I believe it takes away from the contribution of the study by mudding the study's focus. I know that the authors did take out some of this emphasis by removing this from the title but in the response to one of reviewer 1 comments the authors mention the following: "Given that cool surface-based solutions for mitigation of heat pollution are being applied at the level of individual neighborhoods and even then, disadvantaged neighborhoods would only be able to apply them to a part of their surfaces, guidance is needed to determine an optimal strategy that maximizes benefits with a partial application of cool surfaces. There is however, no examination of this problem in the literature, leaving cities and neighborhoods unable to determine how best to make use of limited resources."

The issue of determining an optimal strategy that maximizes the benefits with a partial application is applicable whether it is a disadvantage neighborhood or not. I don't think adding the disadvantaged neighborhood in this paragraph for motivation is needed.

Response: Thank you for the feedback. After careful consideration, we agree with the reviewer that the study is not exclusive to disadvantaged neighborhoods, indeed most cities in the world are unfortunately short of funds and expertise to heat pollution and climate change mitigation. We have removed references to and discussion of disadvantaged neighborhoods and have emphasized that the study is relevant to all cities to make optimal use of their limited resources. For example, we have replaced the relevant discussion in the introduction with the following remarks:

Thus, it is important for urban planners and policymakers to consider solutions for heat mitigation. However, with limited budgets that need to be stretched to provide municipal services, there is a need for solutions and strategies that maximize the benefits from the limited resources available to communities to mitigate heat pollution.

And similarly, we have made modifications to this effect throughout the manuscript, which are marked in the submission.

Comment: With regards to the comment of "Are cool surfaces always reflective?" The authors added the noted paragraph which is a good improvement to address the comment, however I think they should be clearer and not use "cool surfaces" interchangeably with "reflective pavements". I think this is misleading since there could be other cool surfaces that are not reflective.

Response: Thank you for the comment. To avoid any confusion, we have now removed references to "cool surfaces" and simply called them reflective surfaces throughout the manuscript, including in the title.

Reviewer #3:

Comment: For figure 5a and 5b it seems like the colors show that the temperature departure at the building inlet is increase the 2m air temperature relative to Figure 4, which is the control. For 5a and 5b, is the 2m air temperature departure on western buildings <0 or >0. If the 2m temp is greater than zero then there seems to be something wrong with model given the cool surfaces that are placed 100% and upstream respectively for these 2 cases. The red and orange color difference in Figure 5 make it hard to know if it is positive or negative temperature departure. For Figure 5c and 5d the temperature departure away from the cool surface, the temperature departure should be greater than zero.

Response: Thank you for pointing this out, the problem was with the scale of the contours. All departures are either zero or less than zero (this can be double-checked from the corresponding histogram plots), although this can be hard to tell given the orange-red shades are the reviewer pointed out. This means that for this setup of buildings and wind velocity, reflective surfaces either decreased the air temperature or had a negligible effect (this was either very close to the inlet or over parts of the neighborhood with conventional surfaces). None of the departures was greater than zero, which implies that in no case did any reflective pavement strategy do worse than the control case (or to put it succinctly, "something is better than nothing").

We have rescaled the departure contours so that their range is less than or equal to zero (Figures 5, 8, and 11), which is the same as the x-axis in the corresponding histogram plots (Figures 6, 9, and 12 respectively). This should clear up the confusion.

Comment: The reviewer likes Figure 6 (and other similar figures) as an improved way of communicating areas seeing temperature departures relative to cool surfaces. The additional histograms and enhancements to the visual improve the understanding of potential to increase the benefit of cool surfaces

Overall well written and described paper that can assist policymakers, planners, and engineers target cool surface strategies beyond current thinking which is 100% coverage recommendation by product manufacturers.

The B/C also helps to see the upstream placement of the surfaces maximizes benefit in the targeted neighborhood.

Response: Thank you for the positive feedback.

Comment: Authors could add some numbers in conclusions particularly about benefit/cost < 1.0 to cool surfaces not strategically placed.

Response: Thank you for the comment. We have modified the conclusion to add some numbers, though the emphasis is on the principle rather than the details:

While downstream areas experienced the highest heat pollution for different wind directions and aspect ratios, applying reflective surfaces to only that part was less effective relative to cost than applying reflective surfaces to the entire area, with a benefit-to-cost ratio (B/C) less than one for different wind directions and building heights. The benefit relative to cost was about 15-26% *lower* in this case. On the other hand, placing more of the reflective surfaces in the upstream part of the neighborhood yielded disproportionately higher benefits relative to cost (B/C > 1 across wind directions and building heights), as upstream cooling led to cooler air blowing into the downstream part of the neighborhood as well, spreading out the benefits over a larger area. The benefit relative to cost was 15-26% *higher* in this case. From these results, it may be concluded that neighborhoods can maximize the benefit of a limited application of reflective surfaces by concentrating them as close as possible to the upstream part of the neighborhood in the most common direction of wind in the warmest months.

REVIEWERS' COMMENTS

Reviewer #2 (Remarks to the Author):

Thank for addressing my comments.

Response to reviewers

Reviewer #2:

Comment: Thank for addressing my comments.

Response: Thank you for your valuable feedback during the peer review process.